# Differences in Aroma Metabolite Profile, Microstructure, and Rheological Properties of Fermented Milk Using Different Cultures

**DOI:** 10.3390/foods12091875

**Published:** 2023-05-02

**Authors:** Hanh T. H. Nguyen, Mariza Gomes Reis, Yunchao Wa, Renna Alfante, Ryan M. Chanyi, Eric Altermann, Li Day

**Affiliations:** 1AgResearch Ltd., Te Ohu Rangahau Kai, Palmerston North 4474, New Zealand; hanh@nourishing.io (H.T.H.N.); r.chanyi@massey.ac.nz (R.M.C.); e.altermann@massey.ac.nz (E.A.); 2College of Food Science and Engineering, Yangzhou University, Yangzhou 225127, China; 3Riddet Institute, Massey University, Private Bag 11222, Palmerston North 4442, New Zealand; 4School of Veterinary Science, Massey University, Palmerston North 4100, New Zealand

**Keywords:** exopolysaccharide, flavour compounds, microstructure, flow properties

## Abstract

Texture and flavour are the key attributes determining sensory quality and are highly affected by starter cultures. A selection of phenotypic strains is needed to create diverse texture and flavour to meet consumers’ preferences. In this study, the use of five lactic acid bacteria strains in the production of fermented milk, along with the metabolite profiles, microstructure, and rheological properties of the fermented milk samples, was investigated. Our results showed that *Lactobacillus helveticus* (LH) and *Streptococcus thermophilus* (ST) had a stronger acidification during fermentation but resulted in products with a coarser protein network compared to *Lactococcus lactis* (BL1) and *Leuconostoc mesenteroides* (CL3). Milk fermented by LH had the highest viscosity and exopolysaccharide concentration, while milk fermented by ST had the highest concentration of diacetyl. Although *Leuconostoc pseudomesenteroides* (CL3ST) had a minimal acidification capability, it produced high levels of ethyl-derived compounds associated with sweet, fruity, and floral fragrances. The results demonstrated that LH and ST could be used as starter cultures targeting fermented milks with different viscosities, while BL1, CL3, and CL3ST are suitable as adjunct cultures to impact different acidic sharpness and flavour notes.

## 1. Introduction

Fermented dairy products have long been consumed due to their beneficial health effects, extended shelf life, and unique flavours and textures that suit different consumer choices. Starter cultures play an important role in the quality of the products and in delivering the functional attributes that are associated with the structural changes and metabolites generated as part of the fermentation process. In yoghurt production, lactic acid bacteria (LAB) starter cultures containing a mixture of *Streptococcus thermophilus* and *Lactobacillus delbrueckii* subsp. *bulgaricus* are often used. Other adjunct cultures are also employed to impart specific properties to the final products, such as health benefits (probiotics) or sensory properties [1,2,3]. Taste preferences are known to be influenced by environmental and dietary factors that consumers from different ethnic backgrounds experience [4]. Therefore, new micro-organisms that can impact on the structure and bring new flavour attributes to fermented products are highly sought after. In order to develop new technologies to enhance the phenotypic traits of microbes, the metabolic products produced by the microbial fermentation that impart different textures and flavours need to be established.

LAB can be divided into more than 20 genera, among which the most commonly used in dairy fermentation are *Streptococcus, Lactococcus, Lactobacillus, Pediococcus, and Leuconostoc* [5]. Within these genera, the strains belonging to *S. thermophilus, L. helveticus, L. lactis*, and *L. mesenteroides* have been widely used in dairy fermentation for a variety of products, such as yoghurt, kefir, and cheese, while *L. Pseudomesenteroides* has been used to a lesser extent [1,3,6]. The choice of cultures is highly dependent on the attributes of the cultures and the targeted dairy product properties. 

The most important technological properties that often determine the selection of the cultures in the production of a targeted fermented dairy product are acidification capabilities, exopolysaccharide (EPS) production, and the ability to produce metabolites—including aroma-related volatiles. Strong acidification is often a desirable property of a starter culture in dairy production, as this shortens the fermentation time, which can be beneficial for industrial yoghurt production. Cultures with high EPS production capability are often highly sought after, as they can improve the rheological properties, texture, and mouthfeel of the products. Strains with low EPS production may be more suitable for fermented milk with low viscosity and a thin body that is easily suckled. The production of aromatic compounds is also an important attribute of LAB, as this is a powerful tool to create fermented products with different natural and unique flavours to impart sensory properties. 

*S. thermophilus* is a major dairy starter culture in yoghurt production. Several *S. thermophilus* strains have been reported to have fast acidification and high exopolysaccharide (EPS) production [7,8,9]. Han et al. (2022) isolated and characterised 100 *S. thermophilus* strains from 80 naturally fermented dairy products, and they found that 27 had fast acidification and grew independently regardless of the milk type and, hence, could be used as starter cultures; 64 produced EPS, resulting in a smooth and thick fermented milk [7]. Some *S. thermophilus* strains are also able to produce aroma-related volatile compounds. Liu et al. (2022) identified a total of 72 volatile compounds when investigating the milk fermented by 46 *S. thermophilus* strains that could impart the milky, cheesy, creamy, and “fermented related” flavour [9]. Among the 72 volatiles identified, the six predominant aromatic compounds were 2,3-butanedione, 2,3-pentanedione, 2-undecanone, octanoic acid, hexanoic acid, and 2-hydroxy-3-pentanone [9].

*L. helveticus* strains have been used as both starter and adjunct cultures in yoghurt and cheese production due to their high proteolytic properties and generation of flavour-related compounds [3,10,11]. *L. helveticus* H9 was used in yoghurt fermentation and was able to produce a variety of volatile compounds, such as benzaldehyde and acetoin, as well as bioactive peptides such as isoleucyl-prolyl-proline and valyl-prolyl-proline [2]. *L. helveticus* produced compounds that are associated with a strong nutty flavour and reduced bitterness in aged cheese, due to the further breakdown of the bitter peptides caused by its strong proteolytic enzyme system [12,13]. *L. helveticus* ANA12a was reported to produce EPS and to be involved in the regulation of the gut microbiota [14]. 

*L. lactis* is a common mesophilic culture that is widely used in the production of fermented dairy products. *L. lactis* typically has an optimal growth temperature of 30 °C, but through adaptive evolution, the mutant strain TM29 could grow well up to 39 °C [15]. *L. lactis* strains exhibit significant differences in their acidifying and flavour-associated compound production capabilities. Among 227 *L. lactis* strains investigated, 18 strains completed the fermentation process (when milk reached a pH of 4.5) within 12 h, and 2 strains completed the process at 7.6 h, equivalent to that of the commercial culture Chr. Hansen R-704, while there were also 37 strains that could not complete the fermentation process even after 32 h [1]. Strains IAMU11823 and IMAU11919 were extinguished in their ability to produce high contents of 3-methyl butanal and 3-methyl-2-butanone, which are generally known to impart malty and nutty flavours [1]. In addition to these attributes, some strains of *L. lactis* could also produce EPS [16]. Milk fermented by the strain LL-1, which produced ropy and cell-bound EPS, resulted in an improved gel firmness and water retention, likely linked with the large hydrodynamic radius of the EPS produced by this strain [16]. 

*Leuconostoc* spp. are heterofermentative mesophilic LAB, often used as adjunct cultures together with other acid-producing bacteria in dairy processing [17,18]. Several strains of *Leuconostoc* originating from raw milk, cheese, and cheese derivatives have the ability to induce the production of several flavour compounds, such as mannitol, diacetyl, acetoin, and acetate [17,19,20]. Previous studies have mostly focused on the role of *Leuconostoc* spp. in the development of flavour and aroma in cheese products. The effect of *Leuconostoc* spp. on the structure and rheological properties of fermented dairy products is poorly understood, even though some strains of *L. mesenteroides* and *L. pseudomesenteroides* have been found to have the ability to produce EPS [21,22,23].

Fermented milk is one of the most common dairy products, particularly in Asian and Chinese markets. However, there are limited studies about the impact of cultures on the physicochemical properties and aroma metabolite profiles of drinking fermented milk products. Previous studies have so far mostly focused on the technological properties (e.g., acidification, network development, EPS production), the physicochemical properties (e.g., microstructure, rheological properties), or the volatile profiles of the fermented milk as impacted by the cultures. There have been few studies wherein all of these properties were simultaneously investigated. This study provides a more comprehensive characterisation of the five LAB strains and expands our current understanding of the phenotypic dataset of the LAB cultures. 

The aim of this work was to investigate the phenotypic and technological properties of the five cultures from different LAB genera of *S. thermophilus*, *L. helveticus*, *L. lactis*, *L. mesenteroides*, and *L. pseudomesenteroides,* along with their impacts on the fermented milk products (e.g., acidification, gelation, microstructure, rheological properties, volatile metabolites). These five bacterial strains were selected based on the recommendations of dairy industry partners and the initial data available, which showed potential differences in their technological properties and applications. Three of the five strains were newly isolated from NZ local biological sources and have not been reported elsewhere, and so it would be novel to investigate their phenotypic traits, while the other two were of commercial value. The two commercial cultures—Chr. Hansen CH-1 and YF-L811—were also included for reference comparison, with CH-1 marketed for the production of fermented milk with a strong flavour and low viscosity, and YF-L811 for the production of fermented milk with a mild flavour and low viscosity. 

It is well known that LAB isolated from different sources would normally have different distinctive phenotypic traits. As such, we hypothesised that the five LAB strains selected in this study would show different attributes compared to other reported studies on different isolates. This work adds new knowledge for the understanding of the distinctive features of the new isolates, and to identify suitable cultures to meet the growing consumer demand for products with a wider variety of flavours and mouthfeels. The results obtained in this study will set a benchmark for the further enhancement of the functional traits of selected targets [24], as well as enabling the development of consortia for the production of fermented dairy products with improved texture and flavour, tailored for specific consumer markets.

## 2. Materials and Methods

### 2.1. Preparation of Fermented Milk

Five different bacterial cultures were used for milk fermentation individually. *S. thermophilus* (ST) and *L. helveticus* (LH) were kindly provided by Fonterra Research and Development Centre (Palmerston North, NZ). *L. lactis* (BL1), *L. mesenteroides* (CL3), and *L. pseudomesenteroides* (CL3ST) were isolates from New Zealand sheep milk kindly provided by Spring Sheep Dairy New Zealand (Auckland, New Zealand). Two commercial thermophilic culture consortia—CH-1 and YF-L811 (Chr. Hansen)—were used as references to allow the comparison to the real/practical common yoghurt products in an industrial setting. Both commercial samples contained a mixture of *L. delbrueckii* subsp. *bulgaricus* and *S. thermophilus*. 

Cow skimmed milk powder, provided by Fonterra Pty Ltd. (Palmerston North, New Zealand), was rehydrated and kept overnight at 4 °C. The resultant reconstituted milk solution had a composition of 3.0% protein, 0.1% fat, 5.2% lactose, and 9.1% solids, as determined using a MilkoScan FT1 milk analyser (FOSS, Hilleroed, Denmark). The milk solution was heated to 85 °C and tempered at this temperature for 30 min using a shaking water bath (Grant Instruments OLS Aquapro, Cambridge, UK), before being cooled down and inoculated with either one of the commercial starter culture mixes or one of the single experimental bacterial cultures. The freeze-dried direct vat (FDV) commercial starter cultures CH-1 and YF-L811 were used at an inoculation rate of 110 mg/L, while the five experimental bacterial cultures were used at 5% inocula of concentrated stock culture (8.8 × 10^8^–9.1 × 10^9^ CFU/mL), based on previous screening tests and a previous study [25]. The inoculated milk was then fermented at (i) 30 °C for BL1, CL3, and CL3ST, and (ii) 43 °C for ST, LH, YF-L811, and CH-1. When the milk dropped to a pH of 4.5 (or to a maximum fermentation time of 18 h in the event that a pH of 4.5 could not be obtained), the fermentation was terminated by immediate submersion into an ice-water bath for 10 min. The chilled fermented milk was then homogenised at 9500 rpm for 30 s using an Ultra Turrax homogeniser (T25, Ika, Selangor, Malaysia), aliquoted into storage containers, and stored at 4 °C until further analyses.

### 2.2. pH Measurement

Changes in pH during milk fermentation were monitored automatically using a pH titrator (Excellent T7 Titrator, Mettler Toledo, Melbourne, Australia) or a manual pH meter (Orion Star, Thermo Scientific, Auckland, New Zealand). The pH values were recorded at 15 min intervals (when using the automatic pH titrator) or at 1 h intervals (when using the manual pH meter), until the milk reached a pH of 4.5.

### 2.3. Rheological Analysis

The rheological properties of the fermented milk samples after one day of storage were determined using a method described in a previous study [26], with some modifications. The measurements were carried out using a TA DHR-2 rheometer (TA instruments Ltd., New Castle, DE, USA) fitted with a cone and plate (40 mm diameter, 2° angle). The samples were gently mixed by inversion a few times prior to loading onto the plate geometry. After the samples reached 20 °C, they were pre-sheared at 500 s^−1^ for 60 s and then equilibrated for 300 s prior to measurement. The viscosity of the samples was measured as a function of shear rate to generate flow curves over an increasing shear rate range, from 0.1 to 100 s^−1^ in 300 s (upward curve), followed by holding at 100 s^−1^ for 5 s, before decreasing from 100 to 0.1 s^−1^ in 300 s (downward curve). The apparent viscosity at 50 s^−1^ was selected to present, as this viscosity has been reported to be linked with the mouthfeel perception [27]. 

The rheological properties were analysed using the data analysis function available in the TRIOS software (TA instruments Ltd.). The Herschel–Bulkley model—given by the equation σ = σ◦ + K.γ^n^, where σ is the shear stress (Pa), σ◦ is the yield stress (Pa), γ is the shear rate (s^−1^), K is the consistency coefficient (Pa.s^n^), and n is the flow behaviour index—gave the best fit (as indicated by the highest R^2^ values compared to other rheological models for most of the samples (R^2^ > 0.96)). Furthermore, to enable the comparison of our results with those of previous studies, the power-law model (σ = K.γ^n^) was also presented. Yield stress σ◦ is defined as the stress/force required to overcome the material’s inner structural forces to initiate material flow. The consistency coefficient (K) is defined as the viscosity of the material when n = 1 (Newtonian fluid). The flow behaviour index (n) measures the degree of deviation from a Newtonian fluid. The hysteresis loop area (i.e., the area between the upward curve and the downward curve), which indicates the structural breakdown and regeneration (i.e., the degree of thixotropy), was calculated using the TRIOS software.

### 2.4. Microstructural Analysis

Microstructural analysis of the fermented milk samples was performed using an in-house-developed and -optimised confocal laser scanning microscopy (CLSM) method. Briefly, a 50 mL sample was pipetted and placed onto a 35 mm glass-bottomed Petri dish (P35G-1.5-14-C, MatTek, Ashland, MA, USA). An aliquot (1 mL) of a mixture of Fast Green FCF (Sigma-Aldrich, Christchurch, New Zealand) and wheat germ agglutinin Alexa Fluor 488 (WGA488; Invitrogen, Auckland, New Zealand), freshly prepared by mixing Fast Green FCF (1 mg/mL) with WGA488 (1 mg/mL) at a ratio of 1:1 (*v*:*v*), was deposited on top of the sample. The sample was covered by the dish lid and incubated under dark conditions, on ice, for 30 min. The microstructure of the sample was observed using an inverted CLSM (Fluoview FV10i, Olympus, Auckland, New Zealand) with excitation/emission wavelengths set at 635 nm/660–710 nm for Fast Green FCF and 499 nm/520 nm for WGA488. A minimum of nine images were taken for each sample, and representative images are presented.

### 2.5. Particle Size Distribution

The size distribution of fermented milk particles was determined using a Mastersizer 2000 (Malvern Instruments, Malvern, UK), as previously described [25]. An absorption of 0.001 and refractive indices of 1.46 and 1.33 were used for fermented milk particles and water, respectively. The yoghurt samples were gradually added into a circulating cell until an obscuration between 10% and 15% was obtained. The volume-weighted mean diameter D[4,3] and the diameters dv(0,1), dv(0,5), and dv(0,9)—defined as the diameters below which 10%, 50%, and 90% of the volumes of particles are found, respectively—were recorded.

### 2.6. Whey Separation

The whey separation of the fermented milk samples was determined using a previously described method [26]. Briefly, samples contained in the 50 mL centrifuge tubes (Nunc, Thermo Fisher, Auckland, New Zealand) were subjected to light centrifugation (700 g at 8 °C for 10 min). Whey separation was calculated using the equation whey (%) = m2/m1 × 100%, wherein m2 is the weight of expelled whey and m1 is the initial weight of the yoghurt sample. Two technical replicates were performed for each sample in each trial, and three trials of drinking yoghurt were carried out for each bacterial starter culture, resulting in a total of six measurements.

### 2.7. Exopolysaccharide Production

EPS in the fermented milk samples was isolated using a previously described method [8]. Briefly, samples were adjusted to pH 8, heated at 95 °C for 15 min, cooled down to 55 °C, and proteinase K was added (50 U/mL, Macherey-Nagel GmbH & Co. KG, Düren, Germany) and incubated at 55 °C for 16 h to hydrolyse protein. After incubation, absolute ethanol at a ratio of 3:1 (*v*/*v*) was added to the mixture and kept at 4 °C overnight to precipitate EPS, followed by centrifugation (12,000 g at 4 °C for 15 min). EPS pellets were collected and dissolved in 5 mL of Milli-Q water, with 1.73 mL of 80% (*w*/*v*) trichloroacetic acid added, and kept at 4 °C for 8 h. The supernatant was collected after centrifugation (12,000× *g* at 4 °C for 15 min) and diluted with 3× volumes of absolute ethanol, incubated at 4 °C overnight, and then subjected to another centrifugation. After centrifugation, the pellet was dissolved in Milli-Q water and dialyzed for 48 h using an 8–12 kDa cutoff membrane. After dialysis, the samples were transferred to 2 mL centrifuge tubes, and the insoluble material was removed by centrifugation (10,000× *g* for 10 min). The EPS concentration was determined using the phenol–sulphuric acid method with glucose as a standard, and the absorbance of the mixture was measured at 480 nm using a Thermo Scientific Varioskan^TM^ LUX microplate reader ( Thermo Fisher, Franklin, MA, USA).

### 2.8. Analysis of Aromatic Metabolites

The aromatic metabolites in the fermented milk samples were analysed using headspace solid-phase microextraction–gas chromatography–mass spectrometry (SPME-GC-MS). Samples were analysed in daily batches, with 7 samples in each batch. For each treatment, three biological replicates were analysed individually. For each analysis, 5 ± 0.001 g of fermented milk and 125 ng of internal standard (5 µL, 2,3-dichlorobenzene 25 mg/L) were placed into a 20 mL SPME vial. The samples were kept at 4 °C until analysis. To minimise the chance of volatile compound changes, samples were held in the autosampler tray for no longer than 4 h. To minimise analysis sequence effects, the samples were analysed in a random order, and blanks (empty vials) were run every day to check for background signals. Each vial was equilibrated at 55 °C for 10 min under agitation (250 rpm) in the automated sample preparation unit (AOC-6000, Shimadzu, Tokyo, Japan). Following equilibration, a 50/30 μm divinylbenzene-Carboxen-polydimethylsiloxane fibre (Supelco, Bellefonte, PA, USA) was then exposed to the headspace of the sample for 40 min at 55 °C under agitation (250 rpm). After extraction, the volatile compounds were desorbed by direct insertion of the SPME fibre into the injection port of the GC, which was operated in splitless mode at 250 °C for 1 min, followed by a 7 min purge. Headspace volatile compounds were analysed using a GC system (Nexis GC-2030, Shimadzu, Japan) coupled with a mass spectrometer (MS QP 2020 NX, Shimadzu, Japan). The GC system was equipped with a DB-Wax column (30 m, 0.25 mm i.d., and 0.25 μm film thickness). Helium was used as a carrier gas, at a flow rate of 1 mL/min. The initial temperature program employed was isothermal at 40 °C for 5 min, followed by elevation of the temperature to 220 °C at a rate of 4 °C/min and holding for 5 min at 220 °C. Mass spectra were obtained in the electron impact mode at an ionisation voltage of 70 eV, and data were recorded over a range of m/z 35 to 500. Compound identification was carried out by spectral comparison using the Nist05 Library, supported by calculated retention indices relative to a series of alkanes (C5–C24), compared to the National Institutes of Standards and Technology retention index database. Due to some selected analytes coeluting with other compounds, selected masses (*m*/*z*) were used for integration (see Appendix A in the Appendix A).

### 2.9. Statistical Analysis

Minitab software (V17, Minitab Inc., State College, PA, USA) was used for data analysis. The differences between means were assessed by one-way analysis of variance (ANOVA) and Fisher’s paired comparison test using a significance level of *p* = 0.05. To investigate the correlation between variables, including rheological and functional properties, Pearson’s correlation with a 95% confidence level was used. For the heatmap graph, the data were organised in a matrix where rows correspond to samples and columns to aromatic metabolites. The data were mean-centred and scaled for variance one in a column-wise direction to prevent the differences in scale of aromatic metabolite abundances from affecting the assessment of similarity between samples. The order of columns and rows of the scaled data was determined with hierarchical cluster analysis with Euclidian distance and grouping with complete linkage. The outcome of the cluster analysis is shown on the dendrograms presented on the top and right sides of the graph. The reordered data were used to draw the heatmap, where each cell corresponds to the abundance of the aroma metabolites scaled to variance one. Negative values in the colour scale indicate that the values are below the mean abundance for the corresponding aromatic metabolites, and positive values in the colour scales are higher than the corresponding mean abundance value. The heatmap was created using the R package heatmap (version: 1.0.12), running in R (version 4.0.3).

## 3. Results and Discussion

### 3.1. Changes in pH during Fermentation

The changes in pH during fermentation by different cultures are presented in Figure 1. Thermophilic cultures, including the commercial starter cultures (YF-L811 and CH-1) and the single cultures ST and LH, reached pH 4.5 within 4–8 h. Although BL1 and CL3 were able to acidify milk, a much longer time was required. CL3ST showed a significant lag phase and only lowered the pH of milk to 5.6 after 18 h of fermentation, and no coagulation was observed in this sample. As such, the CL3ST sample was not further characterised for particle size and rheological properties, but was only evaluated for EPS concentration and aroma metabolite profile. The shorter fermentation of ST and LH in comparison with BL1, CL3, and CL3ST is consistent with the findings of previous studies [1,2,7]. *S. thermophilus* has been commonly used as a starter culture in yoghurt production, normally co-culturing with *L. bulgaricus.* It has been shown that when *S. thermophilus* was used alone, the fermentation could be completed within 5–6 h [7], while only a few *L. lactis* strains (2 out of 227) could complete the fermentation process after 7.6 h [1]. These results show that the five cultures have different acidification capabilities, with ST and LH having a stronger acidification capability compared to BL1 and CL3, and with CL3ST having the weakest acidification capability.

### 3.2. EPS Concentration

Significant differences in EPS concentration were found across milk samples fermented with different cultures (*p* < 0.05) (Table 1). For the two commercial cultures, YF-L811 produced a higher level of EPS (19.1 µg/mL) compared with CH-1 (14.1 µg/mL). Among the five single bacterial cultures used, LH and CL3 produced the most EPS (11–12 µg/mL), while CL3ST had the lowest EPS content (2–6 µg/mL).

Cultures with EPS production have been reported to improve the texture and sensory properties of the yoghurt [28]. Several *S. thermophilus* cultures were found to produce EPS, while *L. helveticus, L. lactis*, and *Leuconostoc* spp. cultures have often been known for their ability to produce flavour compounds, rather than EPS production [2,8,29]. The EPS concentration in the ST sample used in this study was lower than those typically reported in the literature for *Streptococcus* cultures, with some strains with high EPS production being able to produce up to 860 µg/mL [30,31]. Generally, a higher concentration of EPS is expected to increase the viscosity of the product. However, the molecular structures of EPS (e.g., composition, size and molecular weight, branching degree, etc.) and the interaction of EPS with the milk protein network could also significantly affect the textural and rheological properties of the fermented milk samples [32,33].

### 3.3. Microstructure

Samples fermented by different cultures showed different microstructures (Figure 2). The microstructure of samples fermented by BL1 and CL3 consisted of a fine and dense protein network with small serum pores, whereas the samples fermented by YF-L811, CH-1, and ST contained a coarse protein network with large protein aggregates and serum pores (Figure 2, Panel A). Among those samples fermented with thermophilic cultures, LH showed a microstructure with slightly denser protein network and smaller, more evenly distributed serum pores compared to others (e.g., YF-L811, CH-1, ST).

The appearance of EPS in the microstructure was also different between samples (Figure 2, Panel A). For the two fermented milk samples produced by commercial starter cultures, more EPS was observed in YF-L811 compared to CH-1, consistent with the higher levels of EPS quantified in these samples (Table 1). For the fermented milk samples produced by the five single cultures, the abundance of EPS was similar between BL1, CL3, and LH. On the other hand, more EPS was observed in ST than in the other samples, even though a much lower EPS concentration was measured in the ST fermented milk sample (Table 1). This was likely due to the differences in the degree of binding to the fluorescent probe WGA488 by the EPS of different compositions and/or molecular structures from the two strains. It is possible that the EPS produced by ST might have more N-acetylglucosamine and N-acetylneuraminic acid residues in the structure, to which the WGA binds specifically [28]. As such, the EPS in the ST sample could be visualised with more fluorescence, despite the lower concentration relative to the levels produced by other strains.

Higher-magnification images (Figure 2, Panel B) further revealed some different features in the interaction of EPS with the protein network in the microstructure between the samples. While most of the EPS was tightly attached or embedded directly in the protein (the orange/yellow areas, indicated by the blue arrows), some EPS could also be loosely attached to the edges of the protein aggregates or present in the serum pores within the protein network (the red areas indicated by the white arrows). The differences in the interactions between proteins and EPS observed here might be due to the different molecular compositions of EPS produced by the different single lactic acid bacterial cultures used in this study.

Microstructure is an important physical property of yoghurt, with a significant impact on the rheological properties and sensory perception of the product [34]. However, previous studies have mostly focused on the microstructure of set yoghurt [25,35]. Very few studies have investigated the microstructure of stirred yoghurt, which is described as a gel network or a suspension consisting of interconnected microgels [34]. To the best of our knowledge, this is the first study examining the microstructure of drinking fermented milk as affected by different single cultures, and it adds new knowledge to our understanding of the functional traits of different cultures for different types of fermented dairy products.

### 3.4. Particle Size Distribution and Whey Separation

Figure 3 shows the particle size distribution of the fermented milk samples produced using different starter cultures. All samples had monomodal (one peak) distribution curves. The peaks occurred at 20–23 µm for thermophilic culture samples—including YF-L811, CH1, ST, and LH—and at ~10 µm for mesophilic culture samples, including BL1 and CL3.

The average weighted-volume diameters D[4,3] were significantly different between yoghurt samples (Table 1). In general, the samples fermented by thermophilic cultures (YF-L811, CH-1, ST, LH) had considerably larger particles than those fermented by mesophilic cultures (BL1 and CL3). The diameter of aggregates in the CH-1 sample was slightly smaller than in YF-L811, while there was no significant difference in the particle size between the two mesophilic cultures (BL1 and CL3). The particle size D[4,3] of yoghurt aggregates determined in this study was within the range previously reported for commercial drinking yoghurt (12–61 µm) [36], but considerably smaller than those previously reported for set-style yoghurt (~ 82 µm) fermented using a mixed culture of thermophilic and mesophilic strains [25].

Whey separation was first assessed by observation of the whey that was naturally expelled on the surface of the samples. There was no visible whey observed on top of the samples using this method of assessment. However, some whey could be observed in the body of the fermented milk samples in the screening tests. As such, a light centrifugation was introduced to evaluate the whey separation of the fermented samples. A significantly higher amount of whey was observed in the samples fermented by the thermophilic starter cultures (YF-L811, CH-1, ST, LH) compared to those fermented by mesophilic cultures (BL1 and CL3) (Table 1). CL3ST did not show any whey using both naturally expelled whey and the light centrifugation method. The different amounts of whey expelled across samples in the centrifugation method are linked to the differences in the particle size of the aggregates in the samples (Table 1). These results—including the lower amount of whey separated and the smaller particles in the BL1 and CL3 samples—are linked with their microstructure, with a finer protein network and smaller serum pores (Figure 2). The larger particles/aggregates in the fermented milk samples would result in a greater gravitational pull under the application of the centrifugal force, leading to a faster and greater sedimentation/phase separation. This greater amount of whey separation in the sample may also indicate a weaker protein network that could not mechanically entrap the whey within its structure under gravitational forces.

### 3.5. Rheological Properties

#### 3.5.1. Viscosity

A shear rate sweep (0.1 to 100 s^−1^) was used to investigate the rheological properties of fermented milk samples (Figure 4A and Appendix A). The viscosity of all fermented samples decreased when the shear rate increased (upward curve), and vice versa—the viscosity increased when the shear rate decreased (downward curve). However, the upward and downward curves were not completely overlapping, with gaps observed between the two curves. These results suggest that all fermented milk samples exhibited a shear-thinning behaviour and were both shear-rate- and time-dependent.

The apparent viscosity at 50 s^−1^, which is commonly used as the approximate viscosity perceived in the mouth [27], was different between samples, ranging from 31 to 91 mPa.s (Table 1). As expected, milk fermented by YF-L811 (the commercial starter culture used as a reference for high-viscosity products) showed a higher average viscosity than the CH-1 sample (the commercial culture used as a reference for low-viscosity products) (74 mPa.s vs. 59 mPa.s, respectively), consistent with the higher EPS content quantified (Table 1) and greater abundance of EPS in the microstructure of YF-L811 compared to CH-1 (Figure 2). Samples fermented by the two mesophilic cultures (BL3 and CL1) had similar viscosities, between the referencing range of YF-L811 and CH-1, whereas the viscosities of the two thermophilic cultures (LH and ST) were outside the reference range, with LH above YF-L811 (91 mPa.s) and ST below CH-1 (31 mPa.s).

The higher viscosity of LH compared to ST is consistent with the significantly higher concentration of EPS measured in LH compared with ST (Table 1). Although more EPS was observed within the microstructure in the ST-fermented sample compared to LH (Figure 2), the lower viscosity in ST may indicate that the EPS produced by ST is different to the EPS produced by LH and, thus, contributes less to the enhancement of viscosity in the fermented milk sample. It should also be noted that several previous studies have demonstrated that in addition to the concentration, other properties of the EPS—such as the EPS types, their interaction with the milk protein network [32], the degree of branching, and the molecular weight—could play critical roles in the viscosity of the samples [30,33,37]. The increased amount of free water in the ST sample—as indicated by the larger unstained area (black areas) (Figure 2)—may also have contributed to the lower viscosity of this sample. The high-shear homogenisation process normally used at the final stage of drinking yoghurt production, as used in this study, generally significantly decreases the viscosity of the fermented milk samples, leading to a less profound effect of EPS on the viscosity of the samples. Therefore, any differences in the viscosity observed between the final fermented milk samples produced using different cultures in this study might have been observed at a significantly larger scale if the samples were not subjected to the homogenisation process, as in the case of set-type yoghurt.

Viscosity and flow properties are important physical attributes of food products, particularly for semi-solid foods. Viscosity refers to the thickness and is defined as the ability of a product to resist deformation. In the context of fermented milk products, it refers to the sliminess or thickness/thinness of the product [38]. The apparent viscosity of the fermented milk samples in our study at 50 s^−1^ (except for CL3ST) was within the range reported in the literature for drinking-type yoghurt, where a large variation from 6 to 500 mPa.s has been reported [39,40]. The differences in the reported viscosity are likely associated with the differences in the formulation of the products, starter cultures, and processing conditions (e.g., fermentation temperature, post-fermentation homogenisation, and/or back extrusion), as well as the methods used for rheological measurements. In our study, the processing conditions and rheological measurement settings were the same for all samples; the differences in the viscosity between samples were therefore attributed to the different properties/behaviours of the starter cultures (e.g., their differences in EPS production capability, acidification rates, etc.).

#### 3.5.2. Flow Behaviour Properties

The flow properties of the fermented milk samples were further investigated by fitting the data of the increasing shear rate sweep curve (shear stress vs. shear rate) (Figure 4B–D) to different flow models. The power-law model has been the most commonly used in previous studies for fermented milk, enabling the comparison of our results with those of previous studies (R^2^ ranging from 0.83 to 0.99). In our study, the Herschel–Bulkley model gave the best fit, with the highest correlation coefficient (R^2^ > 0.96) for all samples. By using this model, the yield stress—defined as the force required to initiate the flow of the samples—could also be determined. Table 1 presents the parameters obtained from both the Herschel–Bulkley and power-law models, including yield stress, consistency coefficient K, flow behaviour index n, and hysteresis loop area. In general, lower flow behaviour index and higher consistency coefficient values were obtained when using the power-law model compared to those obtained using the Herschel–Bulkley model.

The yield stresses of the samples fermented by the starter cultures were different, with the highest average values found for YF-L811 and LH (335–337 mPa), followed by those of BL1, CL3, ST, and CH1 (83–144 mPa). These yield stress values are significantly lower than the values reported for stirred yoghurt (3500–29,000 mPa) [41] using a similar approach for yield stress determination. This is associated with the different gel networks of the different yoghurt types: most of the gel network is disrupted in the drinking yoghurt, while only part of the gel network is disrupted in the stirred yoghurt. The stronger gel network in the stirred yoghurt requires a larger force/stress to initiate the flow of the samples. The differences in the yield stresses of samples fermented by different starter cultures imply potential differences in their stability during storage and transportation, when different external forces/stresses might occur and initiate the flow, causing deformation in the samples.

The flow behaviour index, which refers to the degree of deviation of the samples from the Newtonian materials (n = 1), was <1 for the samples, confirming the non-Newtonian behaviour of all of the samples. All samples exhibited a flow behaviour index in the range of 0.46–0.52 (using the Herschel–Bulkley model) or 0.42–0.47 (using the power-law model). These results were within the range (0.19–0.82) reported for the flow behaviour index of fermented milks in previous studies [42,43].

The consistency coefficient K, which refers to the viscosity of the materials when n = 1, was significantly lower for ST compared to the other samples (using both models). The results obtained by fitting the power-law model to the data resulted in a further differentiation in the consistency coefficient, with LH as the highest, followed by YF-L811 and ST as the lowest, consistent with the trend observed for viscosity in these samples. These consistency coefficient values are within the range (25–8384 mPa.s^n^) previously reported for drinking yoghurt [40].

#### 3.5.3. Structural Breakdown and Recovery

Figure 4B–D show the flow curves (shear stress vs. shear rate) of milk samples fermented by different starter cultures. All samples were characterised by the combination of the clockwise hysteresis loop—wherein the shear stress obtained from the increasing shear rate (upward curve) is larger than the corresponding values of the decreasing shear rate (downward curve)—and the anti-clockwise loop, wherein the shear stress obtained from the downward curve is larger than that from the upward curve. These results suggest that all of the fermented milk samples had mixed behaviours of structural breakdown and build-up (i.e., structural regeneration) upon shearing.

The degrees of structural breakdown (indicated by the clockwise loop area) and structural build-up (indicated by the anti-clockwise loop area) were different between samples. For YF-L811, CH-1, and LH, the larger clockwise loops compared to the anti-clockwise loops were easily observed (Figure 4B,C), whereas these differences were not readily visible in other samples (Figure 4D). Further analysis showed that the final hysteresis area (calculated by subtraction of the area of the anti-clockwise loop from that of the clockwise loop) was >0 for all samples (Table 1), suggesting that all samples had structural breakdown behaviour upon shearing as the dominant property. YF-L811 showed a significantly greater hysteresis area, while BL1 and CL3 exhibited smaller areas compared to the other samples (*p* < 0.05), indicating the different degrees of time-dependence and structural breakdown and recovery capability after deformation between samples.

Flow behaviours, including shear thinning and structural breakdown and recovery, have been widely reported for fermented milk. For fluid-like products such as drinking yoghurt, a certain degree of shear thinning and structural recovery after breakdown behaviours is expected. At zero shear rate/without shear (storage conditions), the viscosity of drinking yoghurt is desired to be high and the structure to be stable to prevent sedimentation and whey separation, whereas during pouring and drinking (when a shear is applied to invoke higher shear rates), a structural breakdown occurs and the viscosity is expected to decrease [38]. The increase in viscosity and the consistency coefficient (by using EPS-producing lactic acid bacteria for yoghurts) could increase the residence time of the compounds in the mouth and the time in contact with the palate and taste receptors, resulting in an increase in taste perception [44]. Similarly, an increased consistency coefficient, achieved by the addition of hydrocolloids (e.g., locus bean gum at 0.1%), was found to have positive effects on the preference of the consumers, likely due to the greater desire for a more viscous product [45].

In addition to physical properties, sensory attributes are also key determinant factors for consumers’ acceptance and preference of the products. As such, to obtain a more comprehensive understanding of these bacterial cultures, the volatile flavour profiles of milk fermented by these cultures were investigated.

### 3.6. Volatile Profile

A total of 50 volatile compounds were detected in the fermented milk samples by headspace-SPME-GC-MS (see Appendix A in the Appendix A). The identified volatile compounds belonged to several chemical classes and were derived from the main constituents of skimmed milk (i.e., protein and carbohydrates). The relationship among the volatile profiles of the different fermented milk samples is illustrated using the heatmap visualisation coupled with the hierarchical cluster analysis (Figure 5). Correlation analysis revealed clear differences in the volatile profiles between groups of fermented milk samples.

In terms of technological importance, the five major yoghurt aroma compounds—acetaldehyde, diacetyl, acetone, acetoin, and 2-butanone [46]—were detected in all fermented milk samples. For the two commercial cultures, CH-1 had a significantly higher concentration of acetaldehyde, which is one of the most dominant volatile compounds in yoghurt, associated with the typical “fresh, green, pungent” aroma note [46], consistent with the fact that this culture is normally used for making yoghurt with a strong flavour (according to the manufacturer’s notes). In contrast, YF-L811 had a lower concentration of acetaldehyde but higher concentrations of 2-butanone (associated with varnish-like, sweet, and fruity flavours), diacetyl (buttery/creamy), and acetone (sweet, fruity) [38], indicating that the milk fermented by this culture could have a creamier mouthfeel and a sweeter note, and less of the typical pungent, yoghurt-like aroma. Indeed, it has been reported that the increased concentration of diacetyl led to fermented milk with an enhanced creamy flavour [29].

Significant differences were found in the volatile compound profiles between ST and LH, particularly with the five major yoghurt aroma compounds. The aroma profile of milk fermented by ST featured a low concentration of acetaldehyde and higher concentrations of diacetyl and acetoin (buttery/creamy), which is similar to the profile of milk fermented by the commercial culture YF-L811. In contrast, the milk fermented by LH had a high concentration of acetaldehyde, but low concentrations of diacetyl and acetoin—a profile similar to that of CH-1. These results suggest that milk fermented by the LH culture is likely to have a more typical pungent, yoghurt-like aroma, while the milk fermented by ST is likely to have a creamier note. In addition, milk fermented by LH also contained high concentrations of 2-heptantone and methional (Figure 5, group X), which are the typical aromatic compounds found in cheese [1].

Milk samples fermented by BL1 and CL3 showed similar aroma metabolite profiles, placed between the milks fermented by the two commercial cultures (CH-1 and YF-L811). Overall, their profiles contained low concentrations of acetaldehyde, 2-butanone, and diacetyl, but high concentrations of acidic compounds (Figure 5, group I), which are likely to contribute to the sour taste.

Milk fermented by CL3ST had low concentrations of the key aroma compounds (e.g., acetaldehyde, diacetyl, acetoin) and acids (e.g., butanoic, hexanoic). However, it produced high levels of ethyl-derived compounds (Figure 5, group II), which are associated with the sweet fruity and floral fragrances, normally observed in beverage fermentation [47] or freshly produced yoghurt [48]. These fruity and floral aroma compounds impart a pleasant flavor and hence, could be used to improve the overall aroma of dairy products by minimizing the acidic sharpness and bitterness in yoghurt and cheese products [49].

The results obtained here showed the different volatile profiles among LAB strains. Previous studies found the key aroma compounds in milk fermented by *S. thermophilus* strains were 2,3-butanedione, 2,3-pentanedione, 2-undecanone, octanoic acid, hexanoic acid and 2-hydroxy-3-pentanone [9]; benzaldehyde and acetoin were found important in *L. helveticus* H9 [2]; 3-methyl butanal and 3-methyl-2-butanone in *L. lactis* IAMU11823 and IMAU11919 [1]; and mannitol in *L. pseudomesenteroides* IMAU:11666 [17]. This could be due to the sources of the specific strains and emphasize the importance of in-depth characterization of each specific strain on their ability to produce potentially distinct flavour profiles in fermented dairy products.

Overall, ST and LH showed a strong acidification capability and, hence, can be used as starter cultures. The milk fermented by ST had a low viscosity, a coarse protein network, a relatively large particle size, and more aromatic compounds related to creamy/buttery taste. Hence, it could be used to create products with a creamy flavour but a less viscous (i.e., thinner) mouthfeel and easy to drink. LH-fermented milk had a relatively high viscosity, with a coarse protein network and a high concentration of acetaldehyde. It could be used in the production of drinkable fermented milk targeting a strong yoghurt aroma, or potentially a new complex flavour (e.g., a cheesy note). BL1 and CL3 had a weak acidification capability but, after an extended fermentation time, they produced fermented milk with a fine protein network with small particle size and high concentrations of acidic compounds, which can contribute to a sour taste but likely a smoother mouthfeel. Due to their weak acidification capability, these strains are better suited as adjunct cultures. Although CL3ST is also weak in its acidification capability and, therefore, can only be used as an adjunct culture, it produced high levels of ethyl-derived compounds that could introduce sweet, fruity, and floral fragrances to fermented milk products better suited to Asian consumers’ preferences [50].

It should also be noted that for practical production of yoghurt/fermented milk products, a combination of multiple cultures is generally used. This approach can lead to a shorter fermentation time (e.g., 4–6 h) (beneficial for both cost implications and safety concerns) but also helps to create fermented products with improved texture and aroma as a result of the combination of advantageous attributes of the multiple cultures. For example, a combination of ST and LH could be considered to create a yoghurt with both a strong typical yoghurt flavour and a creamy/buttery taste, or a combination of ST, LH, and CL3ST could be used to create a yoghurt with a strong yoghurt flavour, creamy taste, and fruity aroma notes.

## 4. Conclusions

The five bacterial cultures exhibited different milk-acidifying capabilities and different effects on the microstructure, physical properties, and chemical properties of the fermented milk. The thermophilic cultures *S. thermophilus* (ST) and *L. helveticus* (LH) could be used as starter cultures, with *L. helveticus* (LH) targeting fermented milk with strong yoghurt/cheesy flavour and high viscosity (i.e., thicker body/mouthfeel), in correlation with its high exopolysaccharide content, while *S. thermophilus* ST could target products with a creamy flavour but low viscosity (i.e., thinner mouthfeel, easy to drink), associated with its low concentration of EPS. *L. lactis* BL1 and *L. mesenteroides* CL3 could be used as adjunct cultures to enhance the acidic sharpness and provide a smoother mouthfeel, while *L. pseudomesenteroides* CL3ST could help to create products with a pleasant, sweet, fruity, and floral profile. These results provide valuable insights for the experimental design of future studies using consortia of cultures, and they could be employed as helpful guidelines for dairy producers in the selection of suitable cultures to create fermented milk products with improved texture and flavour.

## Figures and Tables

**Figure 1 foods-12-01875-f001:**
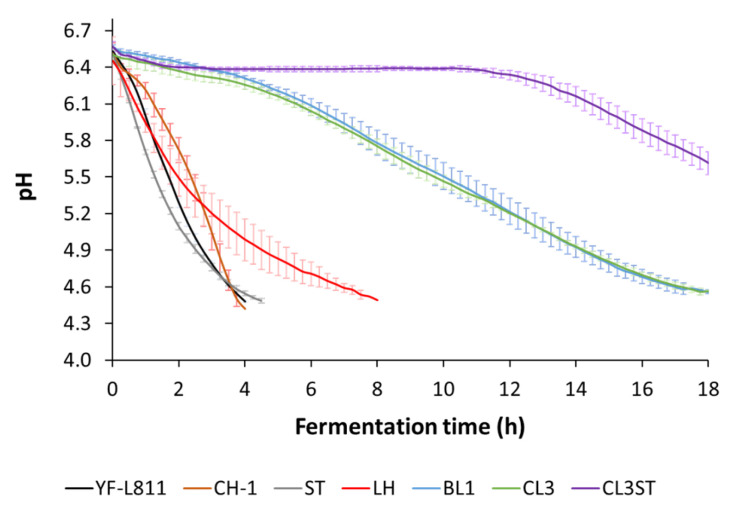
Changes in the pH of milk fermented by commercial starter cultures (YF-L811 and CH1) and five bacterial cultures (LH, ST, BL1, CL3, and CL3ST). Each data point is the average of three measurements (n = 3), and error bars are the standard deviation of the means.

**Figure 2 foods-12-01875-f002:**
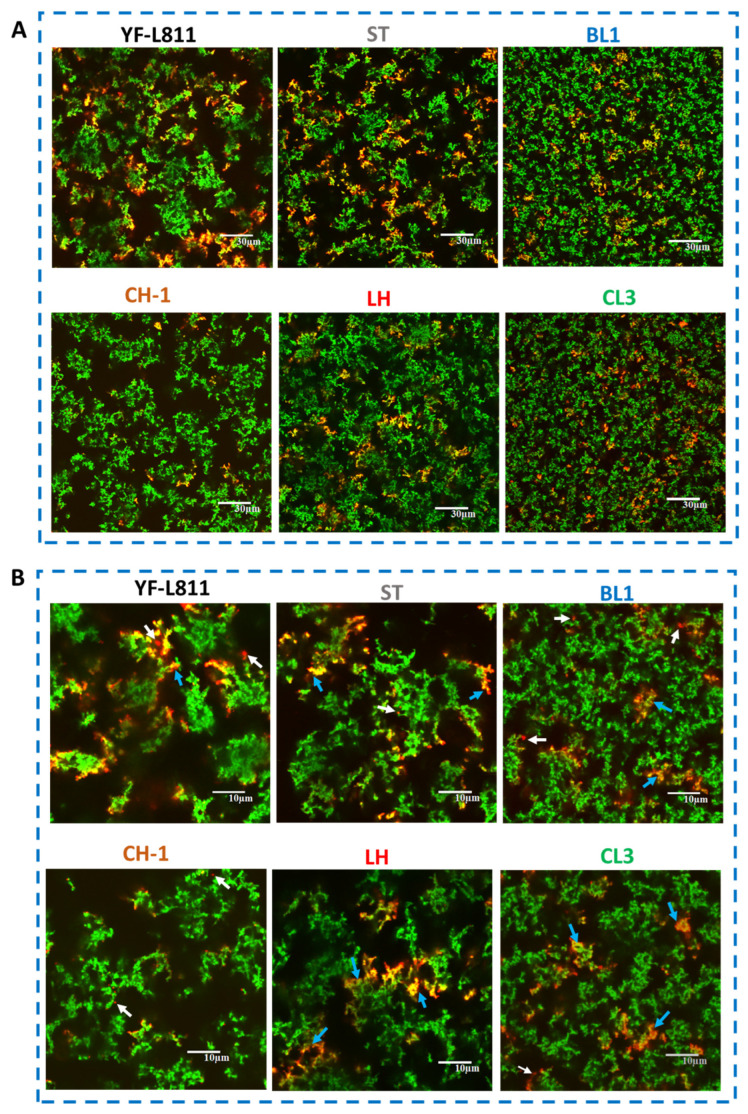
CLSM images showing the microstructure of fermented milk samples using different starter cultures. Protein stained by Fast Green FCF appears as green, and exopolysaccharide EPS stained by WGA488 appears as red. Yellow/orange areas indicate the regions where the EPS is embedded in the protein. Non-stained regions (black areas) indicate serum pores. Blue arrows indicate the EPS tightly attached to the protein network, while white arrows indicate the EPS either in the serum or loosely attached to the protein network. Images were captured using a ×60 objective using a ×1 digital zoom (panel **A**) or ×3 digital zoom (panel **B**). The scale bars are 30 μm in length in (panel **A**) and 10 μm in length in panel B.

**Figure 3 foods-12-01875-f003:**
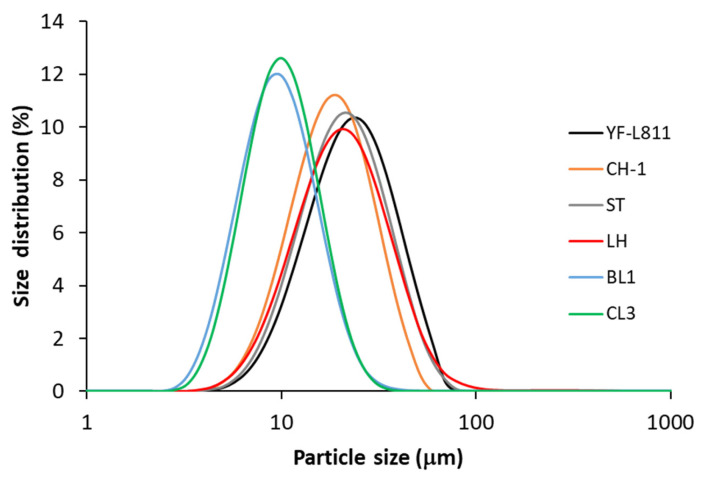
Particle size distribution of fermented milk using different starter cultures. Each data point is the average of six measurements (n = 6).

**Figure 4 foods-12-01875-f004:**
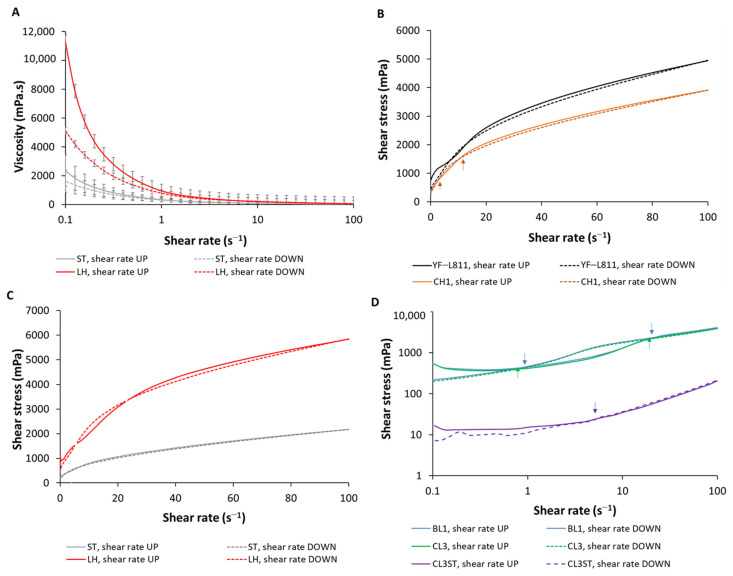
Viscosity (**A**) and shear stress (**B**–**D**) as a function of the shear rate of fermented milk samples. The continuous curves represent the increasing shear rate (upward curves), and discontinuous curves represent the decreasing shear rate (downward curves). In panels (**B**–**D**), the arrows indicate the positions where the upward and downward curves cross over (i.e., transition points between clockwise and anti-clockwise loops). Each data point is the average of six measurements (n = 6).

**Figure 5 foods-12-01875-f005:**
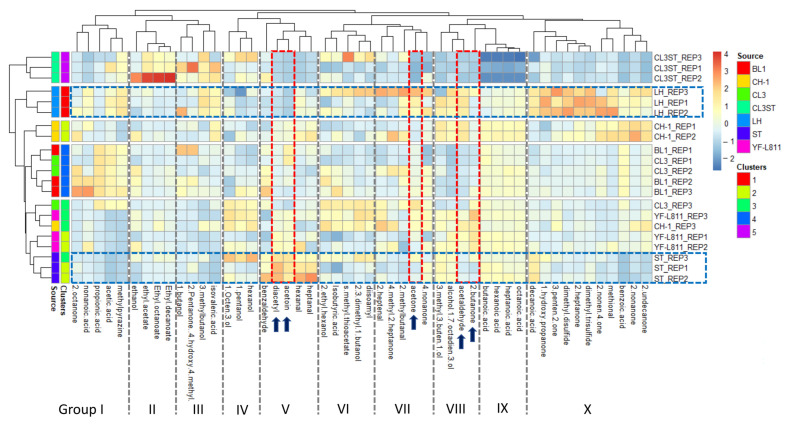
Heatmap showing volatile compounds of milk samples fermented by different cultures. Arrows and the highlighted red box indicate the five key aromatic compounds in yoghurt. According to the heatmap clusters, 10 groups (Group I to X) of volatile metabolites were responsible for the differentiation among milk fermented with different cultures. Most of the groups are formed by volatile metabolites belong to different chemical classes. The exceptions are: Group II that is formed exclusively by ethyl derivative compounds, group V and VII that are formed exclusively by carbonyl compounds (ketones and aldehydes), and group IX that is formed exclusively by carboxylic acids.

**Table 1 foods-12-01875-t001:** EPS concentration, rheological properties, particle size, and whey separation of fermented milk samples. Results were obtained by fitting the increasing shear rate sweep to different rheological models. The results presented are the average of six measurements (n = 6). ^abcde^: Means in the same column with different superscripts are significantly different (*p* < 0.05).

Sample	EPS Concentration (µg/mL)	Viscosity at 50 s^−1^ (mPa.s)	Herschel–Bulkley Model			Power-Law Model		Hysteresis Loop Area (mPa.s^−1^)	Particle Size D[4,3] (µm)	Whey Separation (%)
			Yield Stress σ^o^ (mPa)	Consistency Coefficient K (mPa.s^n^)	Flow Behaviour Index n	Consistency Coefficient K (mPa.s^n^)	Flow Behaviour Index n			
YF-L811	19.1 ± 3.1 ^a^	74 ± 11 ^ab^	335 ± 156 ^a^	430 ± 70 ^a^	0.52 ± 0.04 ^a^	674 ± 103 ^ab^	0.42 ± 0.01 ^b^	6648 ± 1411 ^a^	23.7 ± 0.8 ^a^	61.5 ± 2.0 ^a^
CH-1	14.1 ± 1.3 ^b^	59 ± 10 ^b^	83 ± 84 ^b^	475 ± 92 ^a^	0.46 ± 0.02 ^a^	539 ± 89 ^bc^	0.43 ± 0.01 ^ab^	4601 ± 859 ^ab^	18.6 ± 0.2 ^b^	58.6 ± 2.6 ^ab^
ST	1.8 ± 0.5 ^d^	31 ± 12 ^c^	144 ± 68 ^b^	195 ± 74 ^b^	0.51 ± 0.02 ^a^	315 ± 126 ^c^	0.42 ± 0.01 ^b^	2701 ± 1259 ^bc^	21.9 ± 3.0 ^ab^	57.3 ± 4.7 ^bc^
LH	11.2 ± 2.5 ^b^	91 ± 23 ^a^	337 ± 246 ^a^	517 ± 89 ^a^	0.52 ± 0.04 ^a^	860 ± 272 ^a^	0.43 ± 0.02 ^ab^	4815 ± 1030 ^ab^	22.7 ± 5.2 ^ab^	54.4 ± 1.3 ^c^
BL1	6.1 ± 2.1 ^c^	65 ± 15 ^b^	119 ± 26 ^b^	422 ± 69 ^a^	0.51 ± 0.04 ^a^	510 ± 76 ^bc^	0.47 ± 0.04 ^a^	399 ± 233 ^c^	9.9 ± 1.5 ^c^	30.3 ± 4.3 ^d^
CL3	12.0 ± 2.4 ^b^	61 ± 10 ^b^	99 ± 60^b^	410 ± 106 ^a^	0.50 ± 0.05 ^a^	476 ± 78 ^bc^	0.47 ± 0.02 ^a^	962 ± 580 ^bc^	10.2 ± 1.3 ^c^	37.6 ± 2.8 ^e^
CL3ST	6.0 ± 1.9 ^c^	na	na	na	na	na	na	na	na	na

Note: “na” = not available.

## Data Availability

The data used to support the findings of this study can be made available by the corresponding author upon request.

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
