# Peer review of "Differences in Aroma Metabolite Profile, Microstructure, and Rheological Properties of Fermented Milk Using Different Cultures"

_foods, 2023, doi:10.3390/foods12091875_

Round 1

Reviewer 1 Report

The revision is attached in a separate document

Reviewer 2 Report

This manuscript describes the aroma metabolite profiles, microstructures, rheological properties and size distributions of particle sizes of 7 strains of lactic acid bacteria. First, the authors investigated pH change of media cultured with the bacterial strains. Next, they investigated EPS concentrations of the strains. Then, they investigated the mictostructures of the srains by microscope. They also investigated particle size distributions of the strains. They investigated the viscosities of the strains. Finally they investigated the aroma properties of the strains. Together, they concluded that ST and LH are optimal of starter cultures because of their strong acidification ability and BL1 and CL3 are optimal for adjunct cultures because of their weak acidification ability and strong protein network-forming abilities. CL35T is also optimal for adjunct culture because of its weak acidification ability and strong sweet fruity and floral flagrances-forming ability. On the whole, the manuscript is well written and significant information is presented. Several points should be addressed Reason why these strains were selected should be described Fig4 error bars should be added Comparison with other bacteria should be discussed. you should show the least number of characters before start of the review.

This manucsript describes various characteristics of lactic acid bacteria strains. Experimental details are well described. Data are sufficient.

The reason why the 7 strains were selected should be described well.

Round 2

Reviewer 1 Report

Manuscript was substantially improved by the Authors. I also appreciated responses on all comments